# Microbial Biofertilisers in Plant Production and Resistance: A Review

Domenico Prisa [1,*], Roberto Fresco [2] and Damiano Spagnuolo [3]

1   CREA, Research Centre for Vegetable and Ornamental Crops, Via Dei Fiori 8, 51012 Pescia, Italy
2   CREA, Research Centre for Engineering and Agri-Food Transformation, Council for Agricultural Research and Economics, Via della Pascolare 16, 00016 Monterotondo, Italy; roberto.fresco@crea.gov.it
3   Department of Chemical, Biological, Pharmaceutical and Environmental Sciences, University of Messina, Salita Sperone 31, 98166 Messina, Italy; damiano.spagnuolo@unime.it
*   Correspondence: domenico.prisa@crea.gov.it; Tel.: +39-057-245-1033

**Abstract:** In sustainable agriculture, plant nutrients are the most important elements. Biofertilisers introduce microorganisms that improve the nutrient status of plants and increase their accessibility to crops. To meet the demands of a growing population, it is necessary to produce healthy crops using the right type of fertilisers to provide them with all the key nutrients they need. However, the increasing dependence on chemical fertilisers is destroying the environment and negatively affecting human health. Therefore, it is believed that the use of microbes as bioinoculants, used together with chemical fertilisers, is the best strategy to increase plant growth and soil fertility. In sustainable agriculture, these microbes bring significant benefits to crops. In addition to colonising plant systems (epiphytes, endophytes and rhizospheres), beneficial microbes play a key role in the uptake of nutrients from surrounding ecosystems. Microorganisms, especially fungi, also play a protective function in plants, enhancing the responses of defence systems, and play a key role in situations related to soil iron deficiency or phosphorous solubilisation. Plant-associated microbes can thus promote plant growth regardless of natural and extreme conditions. The most frequently used strategies for growth-promoting microorganisms are nitrogen fixation, the production of growth hormones, siderophores, HCN, various hydrolytic enzymes and the solubilisation of potassium, zinc and phosphorous. Research on biofertilisers has been extensive and available, demonstrating how these microbes can provide crops with sufficient nutrients to increase yields. This review examines in detail the direct and indirect mechanisms of PGPR action and their interactions in plant growth and resistance.

**Keywords:** microbial biofertilisers; microbial symbioses; plant interactions; crop resistance; plant stimulation; sustainable agriculture

## 1. Introduction

Rhizobacteria that support plant growth are known as Plant Growth-Promoting Rhizobacteria (PGPR) [1,2]. The diversity of phenotypic and genotypic characteristics of soil microbiomes makes them complex and difficult to characterise [3]. However, as the rhizosphere has become increasingly important to the bio-sphere in recent years, several PGPRs have been identified that, significantly, have a great impact on plant growth, primarily because they act as an ecological unit [4]. The PGPRs affect plant growth by solubilising insoluble phosphates, fixing atmospheric nitrogen and secreting hormones that control plant growth [5]. Furthermore, through induced systemic resistance (SRI), competition with nutrients, antibiotics, parasitism and the growth suppression of rhizobacteria are mechanisms that lead to increased plant resistance [6]. These communities are very diverse, and their actions can take many forms, including antagonistic action against pathogens in the soil and inducing systemic resistance against pathogens throughout the plant [7].



Plants can be indirectly aided in growing by antagonistic rhizobacteria because they produce various substances that can control pathogens [8]. If the inducing bacteria and the challenging pathogen remain spatially separated, inducing systemic resistance (ISR) can be compared to pathogen-induced acquired systemic resistance (SAR). Different plant species have induced resistance that makes uninfected parts of the plant more resistant to pathogen attacks [9]. The induction of resistance occurs via rhizobacteria either through salicylic acid-dependent SAR pathways or through the bacteria's perception of jasmonic acid and ethylene. Among the many characteristics of rhizobacteria are their antagonistic effects and ability to trigger inflammatory responses. In recent years, many studies have examined the use of PGPR as substitutes for crop protection agents (fertilisers and pesticides) for plant growth promotion [10,11]. Rhizobacteria can alter soil structure, recycle essential elements, decompose organic matter, solubilise mineral nutrients and act as biocontrol agents for soil- and seed-borne pathogens [12–14]. A good understanding of plant growth-promoting rhizobacteria and their interaction with biological and abiotic factors is crucial for bioremediation techniques. This is also relevant for energy generation processes and biotechnological industries such as pharmaceutical, chemical and food industries [15], and rhizobacteria are also useful for reducing the use of chemical fertilisers. The main benefit of this approach is to increase the productivity and sustainability of agricultural systems and soil fertility [16]. The application of fungi, which increase plant defences through biocontrol strategies or can solubilise phosphorus and reduce iron deficiency, is also a strategy currently used in agriculture [16]. As a result, production costs can be reduced and the best soil and crop management practices are identified [17]. The aim of this review was to illustrate the possible benefits of the application of rhizobacteria in plants, the direct and indirect mechanisms they affect, the possible applications of PGPR-based formulations in agriculture, and the prospects for the use of rhizobacteria on crops.

## 2. Plant and Soil Effects of PGPRs

Rhizobacteria that promote plant growth are well known and essential, and this growth enhancement is due to rhizobacteria's characteristics [18]. PGPRs can enhance plant growth and development through various mechanisms [19]. In particular, rhizobacteria produce a variety of substances that alter the entire microbial community in the rhizosphere, and they are capable of supplying nutrients (nitrogen, phosphorus, potassium and essential minerals) or producing plant hormones [20]. For example, the inoculation of rhizobacteria in *Astrophytum* spp. grown in biochar-enriched substrates improves vegetative and root growth and plant flowering (Figure 1) [21]. By acting as biocontrol agents, environmental protectors and root colonisers, PGPRs can also indirectly promote plant growth by reducing the effects of pathogens [22,23]. Sustainable agriculture and plant cultivation can be threatened by the presence of microorganisms, with a deterioration in plant quality and production yields [24]. By fixing nitrogen, mineralising organic compounds, solubilising mineral nutrients and producing phytohormones, PGPRs also facilitate the plant uptake of nutrients and increase resistance to biotic and abiotic stresses. Many species are able to survive particular environmental conditions, such as high temperatures and drought (Table 1) [25]. As an indirect means of achieving soil fertility and plant growth, PGPRs are crucial to a sustainable and ecological approach. This can be achieved through various mechanisms, including antibiotics, HCNs, siderophores and hydrolytic enzymes, and as outlined before, PGPRs can be exploited to decrease the need for agrochemicals such as fertilisers and pesticides and increase soil fertility [26].

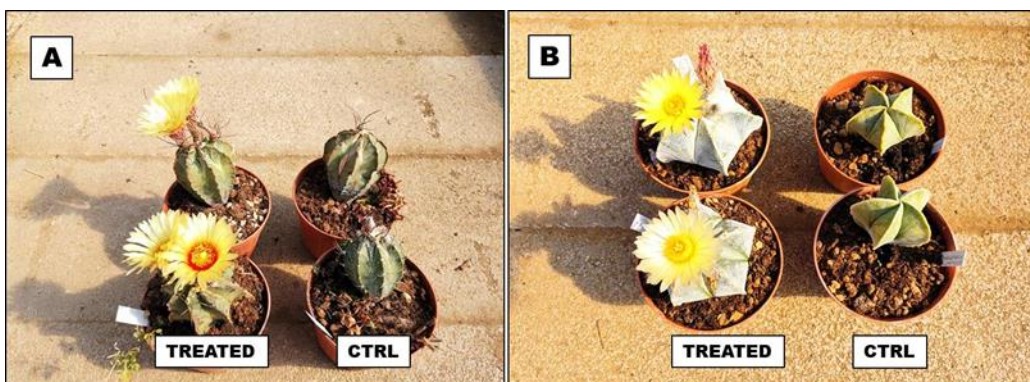

**Figure 1.** Increased vegetative growth and flowering in *Astrophytum capricorne* (**A**) and *Astrophytum myriostigma* (**B**) in plants supplemented with rhizobacteria on biochar substrate [21].

**Table 1.** Bacterial genera and species that are drought resistant [25].

| Bacteria | Crop | Action Mechanism |
|---|---|---|
| *Azospirillum* sp. | Wheat | Highest amounts of N and auxin |
| *Bacillus* sp. | Grass | Responses of antioxidant systems and early proline accumulation |
| *Streptomyces* sp. | Tomato | Increases the content of different sugars |
| *Pseudomonas* sp. | *Arabidopsis* | Higher ACC deaminase activity, gibberellic acid, abscisic acid, indole acetic acid and exopolysaccharide |
| *Enterobacter* sp. | Bean | Enhances proline, malondialdehyde and antioxidant enzymes |
| *Azospirillum brasilense* | Wheat | Lower accumulation of $H_2O_2$ with less enhanced production of proline and activities of catalase and peroxidase |

## 3. Mechanisms Activated Directly by Plant Growth-Promoting Rhizobacteria

In terms of plant growth, phytohormones play a critical role. These are plant hormones that affect the plant's response to its environment. These hormones are produced at one point in the plant and then transferred to another part of the plant, where they are used to promote growth [1]. Roots and leaves grow due to the physical responses caused by these hormones [27]. Some essential plant hormones are auxins, gibberellins, ethylene, cytokinins and abscisic acid [28]. Rhizobacteria produce these phytohormones. In addition to auxins and gibberellins, ethylene, cytokinins and abscisic acid are important phytohormones [29]. Several naturally occurring auxin-like molecules have been described as products of bacterial metabolism in *Azospirillum* sp. cultures. In addition to indole-3-acetic acid (IAA) (between 5 and 50 lg mL$^{-1}$ typically produced according to culture conditions and strain), indol-3-butirric acid (IBA) [30] and phenylacetic acid (PAA) [31], considered in sensu stricto as real auxins, many other indolic compounds (precursors and/or catabolites) have been identified in *Azospirillum* sp. supernatants, including indole-3-lactic acid (ILA), indole-3-ethanol and indole-3-methanol, indole-3-acetamide (IAM) [32], indole-3-acetaldehyde [33], tryptamine (TAM), anthranilate and other uncharacterized indolic compounds [34].

In plant roots and shoots, cytokinins (CKs) play a role in cell division [30]. Among their benefits, there is the growth of cells, the differentiation of cells, apical dominance, axillary bud development and leaf senescence [35,36]. Plants synthesise this hormone, but yeast strains and PGPR strains can also prepare it. In addition, some phytopathogens can synthesise cytokinins. It has been reported that *Azotobacter* species, *Pantoea agglomerans* strains, *Rhizobium* species, *Rhodospirillum rubrum* strains, *Bacillus subtilis* strains, *Pseudomonas fluo-*

*rescens* strains and *Paenibacillus polymyxa* species all produce the cytokinin hormone [37,38]. Some rhizobacteria are able by their actions to mitigate the effects of different types of stress, such as water, salt and heat stress (Table 2) [39]. A class of important plant hormones, gibberellins (GA) control various developmental processes in plants. Their functions include stem elongation, dormancy, germination, flowering and flower development. Several cytokinin-producing polymeric protein receptors synthesise gibberellin, a phytohormone involved in breaking dormancy and other aspects of germination. Gibberellin is the most crucial phytohormone synthesised by some PGPRs. The production and regulation of gibberellin and cytokinin are extremely important [40]. PGPRs and plants produce a variety of phytohormones, including indoloacetic acid. In addition to cell division, other proprieties like gene expression, organogenesis, pigmentation, root development, seed germination, stress resistance, tropical responses and photosynthesis play an essential role in plant cellular responses [41]. Plants and bacteria influence the amount of indole-3-acetic acid (IAA) required to promote plant growth vigorously. The amount of IAA required to promote plant growth depends on the plant and bacterial species. PGPRs produce indole-3-acetic acid, which is responsible for root elongation and the formation of roots. Nearly all plants produce ethylene as a growth hormone, which is key in many physiological changes [42]. Plants respond to biotic and abiotic stresses negatively, affecting root growth and plant growth [43]. The PGPR enzyme 1-aminocyclopropane-1-carboxylate (ACC) deaminase can regulate ethylene production. Inoculation with PGPRs can maintain plant growth and development under stressful conditions, such as drought, salinity, cold and soil pollution, and plants synthesise abscisic acid [25]. This growth hormone activates stress-resistance genes. Abscissic-acid-producing strains, such as *Bacillus licheniformis* Rt4M10, *Azospirillum brasilense* sp. 245 and *Pseudomonas fluorescens* Rt6M10, increase the internal ABA content of plants. As a result, the plants become more resilient to drought. The unavailability of nitrogen can limit plant growth, but phosphorus is also essential for life [44].

**Table 2.** Application of rhizobacteria in mitigating heat stress in plants [35].

| Microbes | Plant | Parameters | Stress |
|----------|-------|------------|--------|
| *Enterobacter SA187* | *Arabidopsis thaliana*, wheat plant | Increased biomass, height, seed weight | High temp. |
| *Septoglomus deserticola* | *Solanum Lycopersicum* | Improved stomatal conductance, water content | Heat drought |
| *Pseudomonas fluorescens, Pantoea agglomerans* | *Triticum aestivum* | Increased antioxidant enzymes | High temp. |
| *B. phytofirmans* | *Solanum tuberosum* | Increased proline and glycine betaine | High temp. |
| *B. cereus* | *Soybean* | Increased chlorophyll and carotenoid | High temp. |

## 4. Microorganisms That Solubilize Phosphate

There are large quantities of phosphate in soil, but they are in an insoluble form that plants cannot utilise for growth since they are insoluble [45]. A group of organisms known as phosphate-solubilizing microorganisms (PSMs) consists of actinobacteria, bacteria, fungi, arbuscular mycorrhizae and cyanobacteria that are capable of hydrolysing organic and inorganic phosphorus into soluble forms, making it available to plants. In Indonesia, Djuuna et al. [46] sampled soil microorganisms, which are commonly associated with the rhizosphere [47]. Agricultural soils with a relevant history of growing vegetables, cereals, and legumes from different regions were collected. The results showed a population of solubilising bacteria ranging between $25 \times 10^3$ and $550 \times 10^3$ CFU $g^{-1}$ of soil and solubilising

fungi between $2.0 \times 10^3$ and $5.0 \times 10^3$ CFU $g^{-1}$ of soil in all areas examined. There is also great diversity in PSM. It is known that bacteria belong to the genera *Azospirillum*, *Bacillus*, *Pseudomonas*, *Nitrosomonas*, *Erwinia*, *Serratia*, *Rhizobium*, *Xanthomonas*, *Enterobacter* and *Pantoea* [47,48]. Non-mycorrhizal fungi include *Penicillium*, *Fusarium*, *Aspergillus*, *Alternaria*, *Helminthosporium*, *Arthrobotrys* and *Trichoderma* [47,48]. *Rhizophagus irregularis* [49,50], *Glomus mossea*, *G. fasciculatum* and *Entrophospora colombiana* are examples of mycorrhizal fungi. PSM occurs in actinobacteria such as *Streptomyces*, *Thermobifida* and *Micrococcus* [51–54], as well as cyanobacteria including *Calothrix braunii*, *Westiellopsis prolifica*, *Anabaena variabilis* and *Scytonema* sp.

## 5. Microbial Activity in Reducing Fe Deficiency

Plants require a small amount of iron from the earth's crust, but Fe deficiency is a nutritional disorder caused by a lack of iron. Plants and microorganisms cannot easily utilize this nutrient in soil because the forms it finds are usually $Fe^{3+}$ oxy-hydroxides. For $Fe^{3+}$ to be readily consumed by plants and microorganisms, it must be reduced to $Fe^{2+}$ [55–57]. Several soil microorganisms have been shown to play a critical role in diminishing Fe deficiency as an environmentally friendly alternative agricultural practice. As well as alleviating biotic and abiotic stresses, these microorganisms have been shown to be beneficial [58,59]. There are rhizobacteria that can colonize the rhizosphere environment, some of which promote nutrient uptake and plant growth; hence, they are referred to as plant growth-promoting rhizobacteria (PGPR) [60,61]. According to their relationship with plant roots, PGPRs fall into two groups: (i) extracellular PGPRs inhabit the rhizosphere, or spaces between root cortex cells, and (ii) intracellular PGPRs inhabit root cells specialized in leguminous nodules [62]. *Micrococcus*, *Pseudomonas*, *Agrobacterium* and *Bacillus* are some of the extracellular PGPR genera. Several studies have shown that PGPR can enhance Fe uptake under limited Fe availability conditions by accumulating and exuding organic acids, phenolic compounds and siderophores and enhancing ferric chelate reductase (FCR) enzyme activities in cucumber [63], *Arabidopsis* [64], pear [65], peach [66] and apple rootstocks [67]. The beneficial effects of PGPR on Fe deficiency have been demonstrated in several studies, but few studies have explored the molecular mechanisms by which PGPR enhances plant Fe uptake. As a result, Zhou et al. [64] and Aras et al. [67] have reported that PGPR activates iron deficiency-related genes like ferric chelate reductase (FRO2) and $Fe^{2+}$ transporter (IRT1).

## 6. Indirect Mechanisms Activated by Plant Growth-Promoting Rhizobacteria

Microorganisms compete for nutrients and colonisation sites in their natural environment fiercely. Various mechanisms of PGPR species have evolved that allow them to reduce competition by releasing antibiotics, lytic enzymes or weak organic acids into their environments (Figure 2) [21,68]. As a result, PGPRs are valuable tools that can be used against plant pathogens. However, there is a possibility of the development of resistant pathogens if antibiotic-producing bacteria are used more frequently. It has been shown that PGPR enzymes secreted by these PGPRs could eliminate pathogens such as *Botrytis cinerea*, *Fusarium oxysporum*, *Sclerotium rolfsii*, *Phytophthora* spp., *Pythium ultimum* and *Rhizoctonia solani* [69,70]. These include cellulases, chitinases, lipases and proteases secreted by the plant. Plants respond to pathogens in two ways: acquired systemic resistance (SAR) and induced systemic resistance (ISR). SAR is implemented in response to a pathogen pre-infection, inducing a hypersensitive reaction, recognisable by a local necrotic lesion of the tissue and an accumulation in the cells of salicylic acid (SA). ISR, on the other hand, induces no visible symptoms and the cells rarely contain SA [71–74]. Systemic acquired resistance is triggered by the infection of a plant by a pathogen. The application of PGPR inocula can induce systemic resistance in the plant, which is useful in protecting against many bacterial pathogens. In addition to promoting fruit growth and ripening, ethylene in plants acts as a phytohormone in response to salt, drought or bacterial pathogens. However, high amounts of ethylene can also cause plant damage [75,76]. This enzyme destroys

1-aminocyclopropane-1-carboxylate, the precursor of ethylene. It relieves plant stress by reducing ethylene levels. Plant root surfaces can be colonised by harmful rhizobacteria that act as biocontrol agents for weeds. They produce toxic compounds known as cyanides, produced by many microorganisms such as bacteria, algae, fungi and plants [77]. Biological weed control agents can be derived from host-specific rhizobacteria, which compete with their counterparts to survive. There is no negative impact on host plants when inoculating with cyanide-producing bacterial strains that produce cyanide [78]. In addition, weed biocontrol agents, such as hydrogen cyanide, are produced, which inhibit the electron transport chain and energy supply to cells. Many harmful microbes compete with PGPRs for nutrients, but these nutrients are present only in trace amounts so that they can limit the disease's causative agent [79]. In fertile soils with abundant non-pathogenic microbes, they colonise plant surfaces quickly and utilise nutrients. These mechanisms can be challenging to study in the system because they inhibit pathogenic microbes from growing. One essential interaction that indirectly supports plant growth is the competition for nutrients between PGPR and pathogens [80].

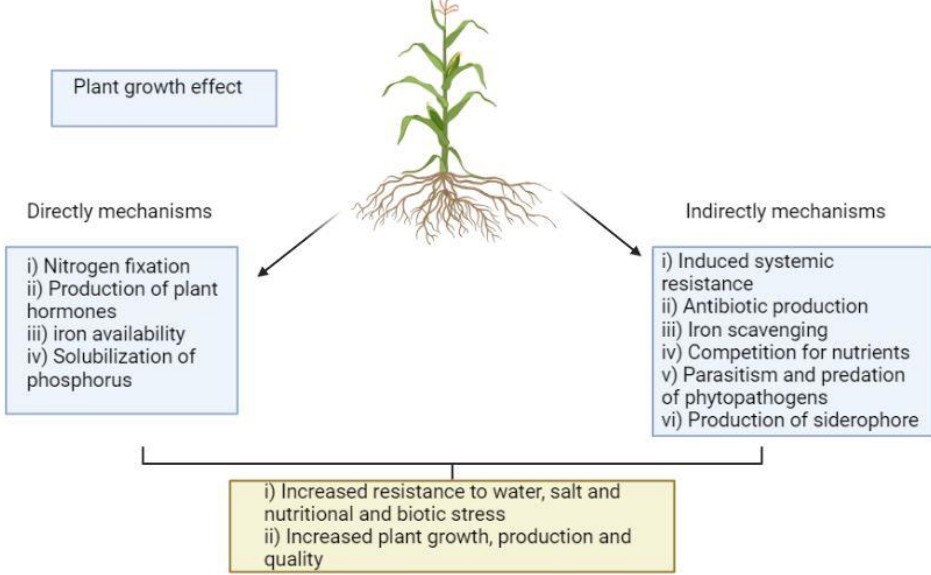

**Figure 2.** PGPB promotes plant growth through the production of siderophores, increasing iron availability and producing hormones such as auxins, gibberellins and cytokinin that modulate the hormone balance of the host plant. The direct mechanisms include biological nitrogen fixation (BNF) via the activity of the nitrogenase enzyme complex, the solubilization of inorganic phosphate in the soil, and the production of siderophores. The indirect mechanisms are attributed to PGPB's occupation of niches and the production of substances that repel phytopathogens and nematodes [21].

## 7. Mechanisms of Biocontrol of Plant Pathogens by Plant Growth-Promoting Rhizobacteria

Known mechanisms of biocontrol are the production of antibiotic substances and degradative enzymes, the production of siderophores, parasitism and predation, competition for space and nutrients, and the induction of systemic resistance in plants (ISR) [81,82]. The antagonistic properties of rhizobacteria on pathogens often occur through the production of a wide variety of antibiotics, the most important of which are 2,4-diacetylphloroglucinol (DAPG), phenazines, pioluteorin, pyrrolnitrin, oligomycin A, kanosomine, zwittermycin A and xanthobaccin, produced by *Pseudomonas*, *Bacillus*, *Streptomyces* and *Stenotrophomonas*. Certain bacteria produce volatile secondary metabolites such as ammonia ($NH_3$) and hydrogen cyanide (HCN), which are effective against various phytopathogenic bacteria and fungi. Siderophores are highly effective chelating agents that bind and transport iron [83]. Chemically, they consist of proteins that represent a selective binding domain for iron. Many microorganisms have developed an iron acquisition strategy based on the production of

siderophores, which are produced precisely when the organism is in an iron-deficient environment [84]. For example, in *Pseudomonas*, there are extremely specific receptors called pyoverdin and pseudobactin, which have a strong affinity for iron. The fluorescence of *Pseudomonas* is due precisely to these siderophores. Fluorescent *Pseudomonas* strains have additional receptors that enable them to obtain iron by taking it away from other phytopathogenic microorganisms living in the soil, thus inhibiting their development [85]. Parasitism occurs when an antagonist is able to live in intimate association with another organism, from which it subtracts all its nutrients. A classic example is bacteriophages, viruses specific to bacteria, which penetrate inside the cell and multiply in large numbers at the expense of the bacterium, which is eventually killed. Conversely, we speak of predation when an organism feeds directly on another organism. A classic example is bacteria of the genus *Bdellovibrio*. These are large bacteria capable of phagocytising other, smaller bacterial cells [86]. Competition for space and nutrients is a biocontrol mechanism that occurs both because of the colonisation of the root and because nutrient compounds and oxygen, which are indispensable for growth, are taken away. Therefore, space and nutrients are taken away from phytopathogenic microorganisms [87]. It is clear that in order to be able to consistently subtract space and nutrients, the micro-organism must grow very rapidly. The induction of systemic resistance in plants (ISR) is a process mediated by the intervention of jasmonic acid and ethylene, involved as signal molecules. ISR is associated with an increase in the sensitivity of plant cells to these hormones. Furthermore, it does not induce the synthesis of pathogenicity-related proteins, except in small quantities, preparing plants to react rapidly and incisively to pathogen attack [88].

## 8. Plant Protection Fungi and Growth Promoters

Defending crops against pathogens and pests is crucial for safeguarding yields and product quality, and intersects with the need to ensure food safety, increase the sustainability of production processes and make efficient use of resources. The availability of healthy, organic agricultural products with minimal use of plant protection products, obtained through production processes that respect both the environment and the safety of operators, is the real challenge for modern agriculture [89]. The concept of biological control stems from the opportunity to counter organisms that are harmful to plants with their own natural enemies, or to their parts and products (extracts, enzymes). Their effectiveness is essentially linked to their high invasive capacity and adaptation to target environments, without leaving residues on the treated crop. The suppressive function is linked to antagonistic interactions [90]. For example, *Coniothyrium minitans*, a mycoparasite of the fungi of the genus *Sclerotinia*, has a terrestrial habitus and draws nourishment solely from the sclerotia of the pathogen, which penetrates directly through the hyphae, making use of the lytic action of the wall structures through specific exoenzymes such as chitinase and glucanase [76]. Another example is *Ampelomyces quisqualis*, a mycoparasite capable of penetrating and producing pycnidia in the vegetative structures of biotrophic pathogenic fungi belonging to the order *Erysiphales*, agents of powdery mildew of grapevine, *Cucurbitaceae, Solanaceae*, strawberry and rose [91]. When different mechanisms of action coexist in the same biocontrol agent, efficacy increases significantly. Endophytic colonisation by non-pathogenic strains of *Fusarium oxysporum* produces biocontrol effects both through increased levels of competition for infection sites on the roots, and through the stimulation of non-specific defence responses in the host; an example is the protection of cucumber from *Pythium ultimum* achieved by root applications of micro-conidial suspensions of the antagonist or in the protection of beans from fusarium blight [92]. Two fungal genera belonging to the family *Hypocreaceae*, *Trichoderma* and *Gliocladium*, comprise numerous species used in broad-spectrum biological control. These fungi, widespread in telluric environments, on wood or other decaying organic matter, reproduce asexually by generating conidia [93]. They grow their hyphae around the host's hyphae and penetrate it, forming appressorium-like structures with cell wall lytic enzymes. The genus *Trichoderma* groups the most commonly formulated species for the biological control of soil-borne

pathogens, such as *Pythium* spp., *Rhizoctonia solani*, *Sclerotium rolfsii*, *Sclerotinia* spp., *Verticillium* spp. and *Fusarium oxysporum*, both on protected and field crops. Fungi of the genus *Trichoderma* release a wide range of antibiotics, enzymes with high antifungal activity and compounds that act as inducers of plant resistance. In aerial applications, *Gliocladium catenulatum* contained alternariasis symptoms on tomato through resistance-inducing mechanisms [94]. There are other antagonistic fungal species with potential commercial development, although they are less common today than those just described. This is the case with *Talaromyces flavus*, proposed for the biological control of certain soil-borne pathogens (*Verticillium dahlie*, *R. solani* and *S. sclerotiorum*), and *Phlebia gigantea*, a biological control agent of root and stem rot in conifers caused by *Heterobasidion* spp. Numerous studies have confirmed the effectiveness of certain microorganisms in promoting crop growth and production, especially when cultivation conditions are sub-optimal (poor soil, presence of biotic and abiotic stresses) [95,96]. The most studied microorganisms in this respect are arbuscular mycorrhizal fungi and fungi belonging to the genus *Trichoderma.* The mycorrhizal fungi establish a symbiosis with the roots of many plants from which they receive energy such as fatty acids and sugars, while the advantage for the plants is that they have a greater availability of water and nutrients. In many cases, the symbiosis with the mycorrhiza also induces a greater growth of the root system, which further improves the absorptive capacity of the crop [97,98]. In addition to the benefits attributed to the symbiosis, the usefulness of mycorrhizae also lies in their ability to favour the structure of soil aggregates, improving their fertility through the solubilisation of various minerals and the production of glomalin, a glycoprotein resistant to degradation [99]. Some species of fungi of the genus *Trichoderma* establish an association with plants through the colonisation of the root surface [100]. The fungus uses the root exudates as nutrients and produces auxin molecules and volatile organic compounds that favour the development of the root system; it also causes an increase in photosynthesis, stomatal conductance, bioavailability and the uptake of nutrients, tolerance to environmental stresses and the growing environment (salinity, low temperatures, heavy metals) in plants [101].

## 9. The Preparation and Application of Commercial Biofertilisers

The use of sustainable technologies to improve plant health has become a necessity due to a number of environmental issues, and biofertilisers play a crucial role in overcoming those issues. In light of this, it has become apparent that biofertilisers are microbes that are vital to sustainable agriculture and play a crucial role in maintaining plant health by acting against pathogens as well as supporting plant growth by providing various nutrients and phytohormones. As a result of the preparation of these formulations, they remain viable while simultaneously enhancing soil fertility and productivity. The formulations are found to increase in number and activity more after being inoculated in the host plant [102]. Biofertiliser formulations that are effective should possess the following desirable characteristics, such as being environmentally friendly, not toxic to the environment and biodegradable. In addition to permitting the addition of nutrients and pH adjustments, they should consist of low-cost raw materials that are readily available and easy to access, should have a long shelf life and should be capable of maintaining metabolically viable high numbers under unfavourable conditions. In addition to liquid biofertilisers, peat-based formulations, granules and freeze-dried powders, there are several types of commercial biofertilisers. Recently developed liquid formulations have gained popularity due to their easy handling and ease of application to seeds and soil [103]. Due to their ease of application compared to conventional solid carrier-based inoculants, liquid biofertilisers offer many advantages. These formulations allow the manufacturer to include adequate amounts of nutrients. In addition, certain inducers can be added to promote the formation of cells, spores or cysts, thus ensuring greater shelf life [104], purity, ease of identification, application and maintenance [105]. Compared to carrier-based powder fertilisers, liquid fertilisers require fewer doses and have a high export potential. Most commercial products contain *Trichoderma* as an active ingredient, and some formulations contain several species

belonging to this genus: *T. asperellum*, *T. gamsii*, *T. viride*, *T. harzianum*. A multitude of commercial proposals, with a predominantly biostimulant function, have a mixed microbiological composition; the association of mycorrhizae of the genus *Glomus* with rhizosphere bacteria (*Bacillus* spp., *Pseudomonas* spp., *Azotobacter* spp., *Azospirillum* spp., *Rhyzobium* spp.) and *Trichoderma* spp. is rather widespread. Other useful fungi sold in mixtures of mycorrhizal inocula belong to the genera *Rhizophagus*, *Clonostachys*, *Arthrobotrys*, *Pochonia* and *Dactylella*, and yeasts of the genus *Pichia*. The combined use of *Trichoderma harzianum* with different strains of *Bacillus subtilis* in repeated pre- and post-transplant treatments can control tomato tracheofusariosis and stimulate both the growth and biosynthesis of vitamin C and lycopene in the berries [106]. These two microorganisms were also combined with a strain of *Pseudomonas fluorescens* and vermicompost, producing the dual effect of reducing tomato tracheofusariasis and increasing antioxidant compounds in the berries [107]. Also, in tomato, the synergistic effect of *Trichoderma* spp. and *Pseudomonas fluorescens* was observed in the biocontrol of bacterial wilt caused by *Ralstonia solanacearum* [108]. The joint use of *Trichoderma*, *Bacillus* and *Pseudomonas*, supported by compost, reduced the incidence of tracheofusariasis in lettuce grown in open fields by up to 69% [109]. The use of composted oak bark both reduced the ability of *Trichoderma* to contain *Phytophthora infestans* in tomato and enhanced the biocontrol efficacy of *Bacillus subtilis* [110]. In potato, the combined treatment of tubers with *Bacillus subtilis* and soil with a mixture of *Trichoderma koningii* and *T. harzianum* controlled *R. solani* and stimulated vegetative plant growth [111].

## 10. Formulated Biofertilisers: Application Methods

Biofertilisers that have been formulated can be applied to soil in a variety of ways, including inoculating seeds with dry fertilizer or liquid fertilizer [112]. The stimulation of plant growth and crop yield by beneficial plant growth-promoting microbiomes either to decrease the use of agrochemicals or pollution caused by them has been assessed in a variety of studies, both in greenhouses as well as in fields. As far as PGPRs go, *Azospirillum* has been evaluated in several studies and is the top choice [113]. *Azospirillum* inoculants can be found in Europe and South Africa, where a number of products, including barley, maize, sorghum and wheat, pre-inoculated with *Azospirillum brasilense*, are already marketed. It is becoming increasingly common for companies to develop new products based on *Azospirillum* and other benefits. The positive results of *Azospirillum* are emphasized, but certain limitations remain to commercialize it, which may be the result of variations in results in field experiments. There are several reasons for the inconsistency of the results, including the physical and chemical conditions of the soil, fluctuations in pH, and the inoculated strain's inability to colonize roots. In addition, fluctuating temperatures and low rainfall during growing may also affect such variable results [114–116]. The support of crop management by beneficial microorganisms is an environmentally friendly alternative to the conventional techniques that are based on chemical inputs, with respect to the increasing consumer expectations of healthy products and current policies towards the implementation of environmentally friendly cropping systems [117]. In addition to biotic stresses, useful microorganisms in agriculture have been shown to increase plant tolerance to abiotic stresses such as flooding, water shortages and excess salinity [118]. Plant growth regulators of microbial origin are of great agrarian and ecological interest, since they offer significant opportunities for eco-friendly agronomic applications. As a result of selected strains, these regulators can also be used in the open field today, thus overcoming certain limitations. It is difficult to colonize the rhizosphere of an adult plant that is already well colonized by resident microorganisms due to high competition [119]. Soil type, temperature, introduced strains, inoculant density and plant species can all influence the immediate response to PGPR soil administration. After inoculation, the introduced population typically drops rapidly, and it is possible that the amount of PGPR colonizing roots will be insufficient to achieve the desired results [120]. Other times, the introduced microorganisms cannot find a free ecological niche in the soil. As well as maintaining the desired character characteristics, the strains used must be capable of surviving the stresses

associated with concentration and stabilisation processes during production. Agricultural crops can be inoculated in a variety of ways:

- Covering the seed at the time of sowing;
- Using confected seeds, i.e., covering with matrices that have included beneficial microorganisms;
- Distributing the product directly in the furrows at the time of sowing;
- Performing covering treatments during plant growth.

Using seed inoculation allows farmers to sow and inoculate at the same time, thus saving time and money. Another option is to encapsulate microbial cells in polymers, particularly alginate, which protect them from environmental stress and allow them to be released into the soil slowly and in large quantities [121]. For example, alginate preparations have been proposed for *Pseudomonas fluorescens* as a biocontrol and biostimulating agent, and for *Azospirillum brasilense* as a biofertilizing and biostimulating agent [122].

The inoculation of fungi can take place via the direct application of spores or mycelium fragments. Numerous formulations are marketed as wettable powders, pastes, creams, water-dispersible microgranules, pellets or liquid preparations. It is essential to comply with the recommended dose and mode of administration stated on the label, and to take into account the expiry date and storage conditions of the product. The presence of chemical residues in the soil and on the crop and the subsequent application of other sterilising treatments may limit the viability and development of beneficial fungi, compromising the effectiveness of the micro-organism treatment. In order to ensure the survival of fungal inocula, enhance saprophytic capacities and encourage the colonisation of the rhizosphere, it is advisable to maintain a temperature and pH range suitable for vegetative development and a good supply of organic matter in pre-biotic soils and to exclude destructive chemical treatments. Treatment is more effective if an initial application is made at the highest dose and repeated applications are made even at lower concentrations; the possibility of increasing the frequency of treatments improves efficacy. Beneficial fungi are used for preventive purposes except in cases where the presence of the pathogen is necessary to allow it to take root and guarantee efficacy. The functionality of the consortium is not always guaranteed by the number of microorganisms; it is essential to seek compatibility and synergies between individuals. For example, in the biocontrol of *Rhizoctonia solani* in beans by evaluating different combinations and inoculation times of *Trichoderma harzianum*, *Rhizophagus intraradices* and *Bacillus pumilus*, it emerged that in simultaneous treatments with substrate infection, the best combination in terms of disease reduction was shown by the *Bacillus–Trichoderma* combination. In prevention, on the other hand, good control was achieved with *Trichoderma* alone, while the combination *T. harzianum–R. intraradices* had no significant effect [123]. For soybean, a consortium consisting of *Trichoderma citrinoviride*, *Pseudomonas aeruginosa*, *Bacillus cereus* and *Bacillus amyloliquefaciens* was tested against *Macrophomina phaseolina* and *Sclerotinia sclerotiorum* [124]. The combination of microorganisms was most active in the production of ammonium, siderophores and lytic enzymes. The consortium consisting of *Trichoderma harzianum*, *Epicoccum* spp., *Bacillus megatherium* and *B. amyloliquefaciens* was successfully employed for the control of black spot in the caryopsis of wheat, caused by the *Cochliobolus sativus* complex, *Alternaria alternata* and *Fusarium graminearum*. In the field, the microbial consortium increased germination and tillering, reduced the incidence of leaf spot and increased seed weight [88].

The combination of *Trichoderma harzianum* and *Pseudomonas fluorescens* had a synergistic effect on the biocontrol of rice bruson, caused by *Magnaporthe oryzae*, and leaf blight due to the bacterium *Xanthomonas orza* pv. *Oryzae* [125]. For tree species, the combination of avirulent strains of *Fusarium oxysporum*, *Phoma* sp. and *Pseudomonas fluorescens* had the ability to reduce the aggressiveness of *Verticillium dahlie* attacks [126].

## 11. The Role of Microbial Biofertilisers in Photosynthesis

Approximately 90% of plant biomass is derived from $CO_2$ assimilation [127], so plant growth depends on the rate of photosynthesis. According to Mia and Shamsuddin [128],

rice plants inoculated with certain strains of *Rhizobia* showed a notable increase in their overall photosynthetic rate. Reactive oxygen species (ROS) are produced as a result of water deficit [129], which damages the photosynthetic apparatus. Under water stress conditions, Heidari and Golpayegani [130] evaluated the effect of *Pseudomonas* sp., *Bacillus lentus* and *Azospirillum brasilensis* on basil plants' photosynthetic capacity and antioxidant activity. Researchers found that these strains decreased water stress by increasing the antioxidant, photosynthetic pigmen, and chlorophyll content of leaves. The effect of inoculating potatoes with *Bacillus* sp. under salt, drought and heavy metal stress was studied by Gururani et al. [131]. It was clear from the study that these bacterial strains influenced the photochemistry of the plants positively, as indicated by the photosynthetic performance indices of inoculated plants. According to Cohen et al. [132], *Azospirillum brasilense* sp. 245 strain was used to inoculate *Arabidopsis thaliana* aba2-1 and Col-0 mutant plants, with morphophysiological and biochemical responses. In addition to other parameters observed, the strain stimulated the formation of photosynthetic and photoprotective pigments. The photosynthetic machinery of the plants was boosted by biofertilisers so that they could grow and survive under stress conditions.

## 12. Biofortification with Microbial Biofertilisers

Micronutrients such as iron, zinc and magnesium are crucial to improving productivity and human health in food crops. A lack of micronutrients in the soil, particularly Zn, is a major limiting factor in achieving maximum yields [133]. In developing countries, cereals are a major source of calories, but they are also low in zinc because they are mostly grown in soils lacking it. Health problems related to zinc deficiency can result from cereal-based diets. The application of microbial biofertilisers can transform poorly available forms of zinc into more available and absorbable forms for plants. There is evidence that most micronutrient deficiencies are associated with wheat and rice, which are dominantly consumed in many countries [133,134]. One strategy that may be effective in enhancing Fe and Zn uptake [133] in the grains is the application of chemical fertilizers, but the disadvantage of using chemical fertilizers is that their micronutrient utilization effectiveness is very small (only 2–5%) [135]. Another advantageous strategy would be to utilize potential microbes for improving the nutrient efficiency of genotypes and fortifying the grains of different crops. The utilization of plant and soil microbiomes to increase micronutrient gaining has been demonstrated in several studies [136]. Microorganisms can significantly improve Fe accumulation in wheat in an efficient and eco-friendly way. Strains of *Bacillus* spp. form spores and are widely explored as plant growth-promoting bacteria (PGPB) in contemporary agriculture for different purposes [137,138]. They secrete siderophores, organic acids and other compounds to promote the uptake of Fe in the rhizosphere of wheat [139,140]. Several *Bacillus* and *Paenibacillus* species increase P, N, K, Fe and Zn [139] contents in maize [141].

## 13. Perspectives on the Use of Microbial Biofertilisers in Agriculture

An essential and safe method for increasing plant growth, resistance to biotic and abiotic stresses and increasing product quality is the use of microbial biofertilisers. In terms of increasing productivity, it is a promising solution [142,143]. In addition, it protects plants from chemicals used to control pests, which can also have a negative impact on the environment. Plant diseases and pests can also be controlled with PGPRs, improving yields. In laboratory and greenhouse experiments, PGPR strains have been advantageous [144,145]. The field of genetic engineering is emerging as a means to improve PGPR strains and explore their potential applications. In addition to all these advances, some environmental barriers and adverse conditions greatly influence the activity of PGPRs [146]. The mixing of strains, the use of improved inoculation techniques and the transfer of the active gene source of antagonists to the host plant can improve the variable efficacy of PGPRs [147]. Furthermore, biocontrol agents need a specific ecological environment to grow and survive, so different conditions may influence their efficacy and use [74,148]. The efficacy of biocontrol agents

can be modified by using compatible mixed inocula in different ecological niches. In addition to these advantages, PGPRs face several challenges. Due to natural variations, it is difficult to predict the behaviour of bacteria in the laboratory and on the farm. These variations can have a significant effect on the entire experiment. Plant type and season can also influence the propagation of PGPRs to recover their viability and biological activity. According to the notion that these bacteria can be applied as biofertilisers in agriculture and forestry, monitoring their activity under stress conditions such as salinity, soil pollution and other environmental conditions that alter crop productivity and yield is essential to understand their applications in different sectors of agriculture. Soil moisture, electrical conductivity and N, P and K concentrations must be monitored under different climatic conditions and bacterial concentrations. This is important in order to develop real, concrete microbial application protocols suitable for different geographical locations.

## 14. Conclusions

There has been substantial progress in the field of using PGP microbes as biofertilisers and biopesticides worldwide. When members of different microbial types interact directly, various key processes occur that ultimately benefit plant growth and soil health. A number of issues will, however, need to be addressed if PGP microbes are extensively utilized. Firstly, moving from the laboratory and greenhouse to field trials will require a number of novel approaches, such as regarding how to grow and store these microbes, as well as the proper facilities for shipping, formulating and applying them. For the widespread use of microbial bio-fertilisers, it would be useful to inform farmers on how best to use them, how to store them, the benefits they can bring to plant cultivation, the possibility of being able to reduce fertilisers and pesticides and the safety for the operator in application. Moreover, plants are exposed to various pathogens that can lead to crop loss and the use of chemical pesticides for fighting diseases, which pose an array of environmental and health problems. In order to feed the emerging population, it is imperative to find alternative strategies that are eco-friendly. In the near future, biofertilisers will not only improve productivity and support the growth of plants during stressful conditions, but also provide a potential alternative strategy for feeding the emerging population. As a result, biofertilisers play a crucial role in modern agriculture, and it is crucial to recognize their importance. Rather than growing plants, sustainable agriculture should cultivate plant–microbial communities, which will ultimately result in high productivity with little energy and chemical investment and minimal environmental impact. In order to achieve sustainable microbial-based agro-technologies, much more effort and collaboration between experts in genetics, molecular biology and ecology are needed.

**Author Contributions:** Conceptualization, D.P.; methodology, writing—original draft preparation D.P. and D.S.; software and investigation, R.F.; writing—review and editing, D.P.; funding acquisition, D.P. All authors have read and agreed to the published version of the manuscript.

**Funding:** This research received no external funding.

**Informed Consent Statement:** Not applicable.

**Data Availability Statement:** All data, tables and figures in this manuscript are original.

**Acknowledgments:** Domenico Prisa would like to express his heartfelt gratitude to his colleagues at CREA Research Centre for Vegetable and Ornamental Crops in Pescia and to all other sources for their cooperation and guidance in writing this article.

**Conflicts of Interest:** The authors declare no conflict of interest.

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
