# Peer review of "Microbial Biofertilisers in Plant Production and Resistance: A Review"

_agriculture, doi:10.3390/agriculture13091666_

Round 1

Reviewer 1 Report

“Microbial biofertilizers in plant production and resistance: A review” discussed Microbial biofertilizers for improving plant growth, soil fertility and resistance. This review topic is interesting and the manuscript is well structured.

With reading the manuscript, I only have the following issues to propose and discuss:

1)    Nice to include Microbial biofertilizers in promoting plant health (to separate microbiome in plant growth or soil fertility);

2)    This review of Microbial biofertilizers mainly focusing on bacteria, how about other microbime for example fungi or protists?

3)    Figure 2 should be improved to highlight PGPB in promoting plant growth through the production of siderophores, increasing iron availability, and producing hormones. In addition, how about phytopathogens of fungi?

Minor editing of English language required

Author Response

Responses to the first reviewer

Good evening I am sending you the changes made according to your suggestions. The article was also checked by an English-speaking scientist.

Microbial biofertilizers in plant production and resistance: A review” discussed Microbial biofertilizers for improving plant growth, soil fertility and resistance. This review topic is interesting and the manuscript is well structured.

With reading the manuscript, I only have the following issues to propose and discuss:

1)    Nice to include Microbial biofertilizers in promoting plant health (to separate microbiome in plant growth or soil fertility);

A section on plant health and biocontrol with related bibliography has also been included as suggested

  1. Mechanisms of biocontrol of plant pathogens by Plant Growth Promoting Rhizo-bacteria

Known mechanisms of biocontrol are the production of antibiotic substances and degradative enzymes, the production of siderophores, parasitism and predation, competition for space and nutrients, and the induction of systemic resistance in plants (ISR) [55,56]. The antagonistic properties of rhizobacteria on pathogens often occur through the production of a wide variety of antibiotics, the most important of which are: 2,4- diacetylphloroglucinol (DAPG), phenazines, pioluteorin, pyrrolnitrin, oligomycin A, kanosomine, zwittermycin A and xanthobaccin, produced by Pseudomonas, Bacillus, Streptomyces and Stenotrophomonas. Certain bacteria produce volatile secondary metabolites such as ammonia (NH3) and hydrogen cyanide (HCN), which are effective against various phytopathogenic bacteria and fungi. Siderophores are highly effective chelating agents that bind and transport iron [57]. Chemically, they consist of proteins that represent a selective binding domain for iron. Many microorganisms have developed an iron acquisition strategy based on the production of siderophores, which are produced precisely when the organism is in an iron-deficient environment [58]. For example, in Pseudomonas there are extremely specific receptors called pyoverdin and pseudobactin, which have a strong affinity for iron. The fluorescence of Pseudomonas is due precisely to these siderophores. Fluorescent Pseudomonas strains have additional receptors that enable them to obtain iron by taking it away from other phytopathogenic microorganisms living in the soil, thus inhibiting their development [59]. Parasitism occurs when an antagonist is able to live in intimate association with another organism, from which it subtracts all its nutrients. A classic example is bacteriophages, viruses specific to bacteria, which penetrate inside the cell and multiply in large numbers at the expense of the bacterium, which is eventually killed. Conversely, we speak of predation when an organism feeds directly on another organism. A classic example is bacteria of the genus Bdellovibrio. These are large bacteria capable of phagocytising other, smaller bacterial cells [60]. Competition for space and nutrients is a biocontrol mechanism that occurs both because of the colonisation of the root and because nutrient compounds and oxygen, which are indispensable for growth, are taken away. Therefore, space and nutrients are taken away from phytopathogenic microorganisms [61]. It is clear that in order to be able to consistently subtract space and nutrients, the micro-organism must grow very rapidly. The induction of systemic resistance in plants (ISR), is a rpocess mediated by the intervention of jasmonic acid and ethylene, involved as signal molecules. SRI is associated with an increase in the sensitivity of plant cells to these hormones. Furthermore, it does not induce the synthesis of pathogenicity-related proteins, except in small quantities, preparing plants to react rapidly and incisively to pathogen attack [62].

References

  1. Hoffland, E.; Hakulinen, J.; Van Pelt, J.A. Comparison of systemic resistance induced by avirulent and nonpathogenic Pseudomonas species. Phytopathology. 1996, 86, 757–762.
  2. Homma, Y.; Sato Z.; Hirayama, F.; Konno, K.; Shirahama, H.; Suzui, T. Production of antibiotics by Pseudomonas cepacia as an agent for biological control of soilborne plant pathogens. Soil biol. Biochem. 1989, 21, 723–728.
  3. Howell, C.R.; Stipanovic, R.D. Suppression of Pythium ultimum induced Damping-off of cotton seedlings by Pseudomonas fluorescens and its antibiotic, pyoluteorin. Phytopathology. 1980, 70, 712–715.
  4. Jenifer, M.R.A.; Reena,A., Aysha, O.S.; Valli, S.; Nirmala, P.; Vinothkumar, P. Isolation of siderophore producing bacteria from rhizosphere soil and their antagonistic activity against selected fungal plant pathogens. Int.J.Curr.Microbiol. Appl. 2013, 2, 59–65.
  5. Larkin, R.P., Tavantzis, S. Use of biocontrol organisms and compost amendments for improved control of soilborne diseases and increased potato production. Am.J.Potato Res. 2013, 90, 157–167.
  6. Mehta, C.M.; Palni, U.; Franke-Whittle, I.H.; Sharma, A.K. Compost: Its role, mechanisms and impact on reducing soil-borne plant diseases. Waste manage. 2014, 34, 607–622.
  7. Ongena, M.; Daayf, F.; Jacques, P.; Thonart, P.; Benhamou, N.; Paulitz, T.C.; Cornelis, P.; Koedam, N.; Belanger, R.R. Protection of cucumber against Pythium root rot by fluorescent pseudomonads: predominant role of induce resistance over siderophores and antibiosis. Plant pathology. 1999, 48, 66–76.
  8. Pal, K.K.; Mc Spadden Gardener, B. Biological control of plant pathogens. The plant Health Instructor. APSnet. 2006, 1–25

2)    This review of Microbial biofertilizers mainly focusing on bacteria, how about other microbime for example fungi or protists?

A section on fungi has been included with the relevant bibliography

  1. Plant protection fungi and growth promoters

Defending crops against pathogens and pests is crucial for safeguarding yields and product quality, and intersects with the need to ensure food safety, increase the sustain-ability of production processes and make efficient use of resources. The availability of healthy, organic and zero-residue agricultural products, obtained through production processes that respect both the environment and the safety of operators, represents the real challenge of modern agriculture [62]. The concept of biological control stems from the opportunity to counter organisms that are harmful to plants with their own natural enemies, or their parts and products (extracts, enzymes). Their effectiveness is essentially linked to their high invasive capacity and adaptation to target environments, without leaving residues on the treated crop. The suppressive function is linked to antagonistic interactions [62]. For example, Coniothyrium minitans, a mycoparasite of the fungi of the genus Sclerotinia, has a telluric habitus and draws nourishment solely from the sclerotia of the pathogen, which penetrates directly through the hyphae, making use of the lytic action of the wall structures by specific exoenzymes such as chitinase and glucanase [62]. Another example is Ampelomyces quisqualis, a mycoparasite capable of penetrating and producing pycnidia in the vegetative structures of biotrophic pathogenic fungi belonging to the order Erysiphales, agents of powdery mildew of grapevine, Cucurbitaceae, Solanaceae, strawberry and rose [63]. When different mechanisms of action coexist in the same biocontrol agent, efficacy increases significantly. Endophytic colonisation by non-pathogenic strains of Fusarium oxysporum produces biocontrol effects both through increased levels of competition for infection sites on the roots, and through stimulation of non-specific defence responses in the host; an example is the protection of cucumber from Pythium ultimum achieved by root applications of micro-conidial suspensions of the antagonist or in the protection of bean from fusarium blight [64]. Two fungal genera be-longing to the family Hypocreaceae, Trichoderma and Gliocladium, comprise numerous species used in broad-spectrum biological control. These fungi, widespread in telluric environments, on wood or other decaying organic matter, reproduce asexually by gen-erating conidia [65]. They grow their hyphae around the host's hyphae and penetrate it, forming appressorium-like structures with cell wall lytic enzymes. The genus Trichoderma groups the most commonly formulated species for the biological control of soil-borne pathogens such as Pythium spp., Rhizoctonia solani, Sclerotium rolfsii, Sclerotinia spp., Ver-ticillium spp., and Fusarium oxysporum, both on protected and field crops. Fungi of the genus Trichoderma release a wide range of antibiotics, enzymes with high antifungal ac-tivity and compounds that act as inducers of plant resistance. In aerial applications, Gli-ocladium catenulatum contained alternariasis symptoms on tomato through resistance inducing mechanisms [66]. There are other antagonistic fungal species with potential commercial development, although they are less common today than those just described. This is the case with Talaromyces flavus, proposed for the biological control of certain soil-borne pathogens (Verticillium dahlie, R. solani and S. sclerotiorum) and Phlebia gigantea, a biological control agent of root and stem rot of conifers caused by Heterobasidion spp. Numerous studies have confirmed the effectiveness of certain microorganisms in promoting crop growth and production, especially when cultivation conditions are sub-optimal (poor soil, presence of biotic and abiotic stresses) [67,68]. The most studied microorganisms in this respect are arbuscular mycorrhizal fungi and fungi belonging to the genus Trichoderma. The mycorrhizal fungi establish a symbiosis with the roots of many plants from which they receive energy such as fatty acids and sugars, while the advantage for the plants is that they have greater availability of water and nutrients. In many cases, the symbiosis with the mycorrhiza also induces a greater growth of the root system, which further improves the absorptive capacity of the crop [69,70]. In addition to the advantages attributed to the symbiosis, the usefulness of mycorrhizae also lies in their ability to promote the structure of soil aggregates, improving their physical fertility, through the production of glomalin, a glycoprotein resistant to degradation [71]. Some species of fungi of the genus Trichoderma establish an association with plants through colonisation of the root surface [72]. The fungus uses the root exudates as nutrients and produces auxin molecules and volatile organic compounds that favour the development of the root system; it also causes an increase in photosynthesis, stomatal conductance, bioavailability and uptake of nutrients, tolerance to environmental stresses and the growing environment (salinity, low temperatures, heavy metals) in plants [73].

References

  1. Pal, K.K.; Mc Spadden Gardener, B. Biological control of plant pathogens. The plant Health Instructor. APSnet. 2006, 1–25
  2. Sztejnberg, A.; Galper, S.; Mazar, S.; Lisker, N. Ampelomyces quisqualis for biological and integrated control of powdery mildews in Israel. Journal of Phytopathology. 1989, 124, 285–295.
  3. Romero, D.; Rivera, M.E.; Cazorla, F.M; De Vicente, A.; Perez-Garcia, A. Effect of mycoparasitic fungi on the development of Sphaerotheca fusca in melon leaves. Mycological Research. 2003, 107, 64–71.
  4. Goettel, M.S.; Koike, M.; Kim, J.J.; Aiuchi, D.; Shinya, R.; Brodeur, J. Potential of Lecanicillium spp. for management of insects, nematodes and plant diseases. Journal of invertebrate Pathology. 2008, 98, 256–261.
  5. Dhingra, O.D.; Coelho-Netto, R.A.; Rodrigues, F.A., Silva, G.J.; Maia, C.B. Selection of endemic nonpathogenic endophytic Fusarium oxysporum from bean roots and rhizosphere competent fluorescent Pseudomonas species to suppress Fusarium-yellow of beans. Biological control. 2006, 39, 75–86.
  6. Singh, A.; Shukla, N.; Kabadwal, B.C.; Tewari, A.K.; Kumar, J. Review on plant Trichoderma pathogen interaction. International journal of current microbiology and applied sciences. 2018, 7, 2382–2397.
  7. Egel, D.S.; Hoagland, L.; Davis, J.; Marchino, C.; Bloomquistc, M. Efficacy of organic disease control products on common foliar diseases of tomato in field and greenhouse trials. Crop protection. 2019, 122, 90–97.
  8. Skinner, M.; Parker, B.L.; Kim, J.S. Role of entomopathogenic fungi in integrated pest management. Integrated Pest Management. 2014, 272–266.
  9. Rouphael, Y.; Franken, P.; Schneider, C.; Schwarzd, D.; Giovannetti, M.; Agnolucci, M.; De Pascale, S.; Bonini, P., Colla, G. Arbuscolar mychorrhizal fungi act as biostimulants in horticultural crops. Scientia Horticulturae. 2015, 196, 91–108.
  10. Mo, Y.; Wang, Y.; Yang, R.; Zheng, J.; Liu, C.; Li, H.; Ma, J.; Zhang, Y.; Wei, C.; Zhang, X. Regulation of plant growth, photosynthesis, antioxidation and osmosis by an arbuscolar mychorrhizal fungus in watermelon seedlings under well-watered and drought conditions. Frontiers in plant Science. 2016, 7, 644.
  11. Colla, G.; Rouphael, Y.; Di Mattia, E.; El-Nakhel, C.; Cardarelli, M. Co-inoculation of Glomus intraradices and Trichoderma atroviride acts as a biostimulant to promote growth, yield and nutrient uptake of vegetable crops. Journal of the Science of Food and Agriculture. 2015, 95, 1706–1715.
  12. Zhang, S.; Gan, Y.; Xu, B. Application of plant growth promoting fungi Trichoderma longibrachiatum T6 enhances tolerance of wheat to salt stress through improvement of antioxidative defense system and gene expression. Frontiers in Plant Science. 2016, 7, 1405.

3)    Figure 2 should be improved to highlight PGPB in promoting plant growth through the production of siderophores, increasing iron availability, and producing hormones. In addition, how about phytopathogens of fungi?

Figure 2 has been suitably modified as suggested.

Reviewer 2 Report

Comments of the manuscript “Microbial biofertilizers in plant production and resistance: A review”   I do not recommend the publication of the manuscript with this structure and information that it presents, for the following reasons that I mention below:   The title of manuscript mentions the term “Microbial biofertilizers”, however, the manuscript is focused on Plant Growth Promoting Rhizobacteria. Microbial biofertilizers should include fungi (i.e., Trichoderma) or/and microalgae. Currently, the role of different microbial groups is highly relevant due to their potential as biofertilizers.   I consider that the sections from numbers 5 to 8 should contribute with more relevant information on the topic of "Microbial biofertilizers". There is a vast literature on this topic and therefore those sections should contribute further information relevant to this Review or Minireview.   Next, I share some references that I consider relevant, and that this manuscript should consider for a better version.

Dasgupta, Kulbhushan Kumar, Rashi Miglani, Rojita Mishra, Amrita Kumari Panda, Satpal Singh Bisht. Chapter 1 - Microbial biofertilizers: Recent trends and future outlook,Editor(s): Surajit De Mandal, Ajit Kumar Passari, Recent Advancement in Microbial Biotechnology, Academic Press,2021,Pages 1-26, https://doi.org/10.1016/B978-0-12-822098-6.00001-X.

Kamini Gautam, Chhavi Sirohi, N. Raju Singh, Yourmila Thakur, Surendra Singh Jatav, Kiran Rana, Manoj Chitara, Rajendra Prasad Meena, Ashish Kumar Singh, Manoj Parihar. Chapter 1 - Microbial biofertilizer: Types, applications, and current challenges for sustainable agricultural production, Editor(s): Amitava Rakshit, Vijay Singh Meena, Manoj Parihar, H.B. Singh, A.K. Singh, Biofertilizers, Woodhead Publishing, 2021,https://doi.org/10.1016/B978-0-12-821667-5.00014-2.

Jupinder Kaur, Vishnu. Chapter 8 - Bacterial inoculants for rhizosphere engineering: Applications, current aspects, and challenges, Editor(s): Ramesh Chandra Dubey, Pankaj Kumar. Rhizosphere Engineering, Academic Press, 2022, Pages 129-150, https://doi.org/10.1016/B978-0-323-89973-4.00004-1.

Author Response

Responses to the second reviewer

Good evening I am sending you the changes made according to your suggestions. The article was also checked by an English-speaking scientist.

Comments of the manuscript “Microbial biofertilizers in plant production and resistance: A review”   I do not recommend the publication of the manuscript with this structure and information that it presents, for the following reasons that I mention below:   The title of manuscript mentions the term “Microbial biofertilizers”, however, the manuscript is focused on Plant Growth Promoting Rhizobacteria. Microbial biofertilizers should include fungi (i.e., Trichoderma) or/and microalgae. Currently, the role of different microbial groups is highly relevant due to their potential as biofertilizers.   I consider that the sections from numbers 5 to 8 should contribute with more relevant information on the topic of "Microbial biofertilizers". There is a vast literature on this topic and therefore those sections should contribute further information relevant to this Review or Minireview.   Next, I share some references that I consider relevant, and that this manuscript should consider for a better version.

Dasgupta, Kulbhushan Kumar, Rashi Miglani, Rojita Mishra, Amrita Kumari Panda, Satpal Singh Bisht. Chapter 1 - Microbial biofertilizers: Recent trends and future outlook,Editor(s): Surajit De Mandal, Ajit Kumar Passari, Recent Advancement in Microbial Biotechnology, Academic Press,2021,Pages 1-26, https://doi.org/10.1016/B978-0-12-822098-6.00001-X.

Kamini Gautam, Chhavi Sirohi, N. Raju Singh, Yourmila Thakur, Surendra Singh Jatav, Kiran Rana, Manoj Chitara, Rajendra Prasad Meena, Ashish Kumar Singh, Manoj Parihar. Chapter 1 - Microbial biofertilizer: Types, applications, and current challenges for sustainable agricultural production, Editor(s): Amitava Rakshit, Vijay Singh Meena, Manoj Parihar, H.B. Singh, A.K. Singh, Biofertilizers, Woodhead Publishing, 2021,https://doi.org/10.1016/B978-0-12-821667-5.00014-2.

Jupinder Kaur, Vishnu. Chapter 8 - Bacterial inoculants for rhizosphere engineering: Applications, current aspects, and challenges, Editor(s): Ramesh Chandra Dubey, Pankaj Kumar. Rhizosphere Engineering, Academic Press, 2022, Pages 129-150, https://doi.org/10.1016/B978-0-323-89973-4.00004-1.

A section on fungi has been included with the relevant bibliography

  1. Plant protection fungi and growth promoters

Defending crops against pathogens and pests is crucial for safeguarding yields and product quality, and intersects with the need to ensure food safety, increase the sustain-ability of production processes and make efficient use of resources. The availability of healthy, organic and zero-residue agricultural products, obtained through production processes that respect both the environment and the safety of operators, represents the real challenge of modern agriculture [62]. The concept of biological control stems from the opportunity to counter organisms that are harmful to plants with their own natural enemies, or their parts and products (extracts, enzymes). Their effectiveness is essentially linked to their high invasive capacity and adaptation to target environments, without leaving residues on the treated crop. The suppressive function is linked to antagonistic interactions [62]. For example, Coniothyrium minitans, a mycoparasite of the fungi of the genus Sclerotinia, has a telluric habitus and draws nourishment solely from the sclerotia of the pathogen, which penetrates directly through the hyphae, making use of the lytic action of the wall structures by specific exoenzymes such as chitinase and glucanase [62]. Another example is Ampelomyces quisqualis, a mycoparasite capable of penetrating and producing pycnidia in the vegetative structures of biotrophic pathogenic fungi belonging to the order Erysiphales, agents of powdery mildew of grapevine, Cucurbitaceae, Solanaceae, strawberry and rose [63]. When different mechanisms of action coexist in the same biocontrol agent, efficacy increases significantly. Endophytic colonisation by non-pathogenic strains of Fusarium oxysporum produces biocontrol effects both through increased levels of competition for infection sites on the roots, and through stimulation of non-specific defence responses in the host; an example is the protection of cucumber from Pythium ultimum achieved by root applications of micro-conidial suspensions of the antagonist or in the protection of bean from fusarium blight [64]. Two fungal genera be-longing to the family Hypocreaceae, Trichoderma and Gliocladium, comprise numerous species used in broad-spectrum biological control. These fungi, widespread in telluric environments, on wood or other decaying organic matter, reproduce asexually by gen-erating conidia [65]. They grow their hyphae around the host's hyphae and penetrate it, forming appressorium-like structures with cell wall lytic enzymes. The genus Trichoderma groups the most commonly formulated species for the biological control of soil-borne pathogens such as Pythium spp., Rhizoctonia solani, Sclerotium rolfsii, Sclerotinia spp., Ver-ticillium spp., and Fusarium oxysporum, both on protected and field crops. Fungi of the genus Trichoderma release a wide range of antibiotics, enzymes with high antifungal ac-tivity and compounds that act as inducers of plant resistance. In aerial applications, Gli-ocladium catenulatum contained alternariasis symptoms on tomato through resistance inducing mechanisms [66]. There are other antagonistic fungal species with potential commercial development, although they are less common today than those just described. This is the case with Talaromyces flavus, proposed for the biological control of certain soil-borne pathogens (Verticillium dahlie, R. solani and S. sclerotiorum) and Phlebia gigantea, a biological control agent of root and stem rot of conifers caused by Heterobasidion spp. Numerous studies have confirmed the effectiveness of certain microorganisms in promoting crop growth and production, especially when cultivation conditions are sub-optimal (poor soil, presence of biotic and abiotic stresses) [67,68]. The most studied microorganisms in this respect are arbuscular mycorrhizal fungi and fungi belonging to the genus Trichoderma. The mycorrhizal fungi establish a symbiosis with the roots of many plants from which they receive energy such as fatty acids and sugars, while the advantage for the plants is that they have greater availability of water and nutrients. In many cases, the symbiosis with the mycorrhiza also induces a greater growth of the root system, which further improves the absorptive capacity of the crop [69,70]. In addition to the advantages attributed to the symbiosis, the usefulness of mycorrhizae also lies in their ability to promote the structure of soil aggregates, improving their physical fertility, through the production of glomalin, a glycoprotein resistant to degradation [71]. Some species of fungi of the genus Trichoderma establish an association with plants through colonisation of the root surface [72]. The fungus uses the root exudates as nutrients and produces auxin molecules and volatile organic compounds that favour the development of the root system; it also causes an increase in photosynthesis, stomatal conductance, bioavailability and uptake of nutrients, tolerance to environmental stresses and the growing environment (salinity, low temperatures, heavy metals) in plants [73].

References

  1. Pal, K.K.; Mc Spadden Gardener, B. Biological control of plant pathogens. The plant Health Instructor. APSnet. 2006, 1–25
  2. Sztejnberg, A.; Galper, S.; Mazar, S.; Lisker, N. Ampelomyces quisqualis for biological and integrated control of powdery mildews in Israel. Journal of Phytopathology. 1989, 124, 285–295.
  3. Romero, D.; Rivera, M.E.; Cazorla, F.M; De Vicente, A.; Perez-Garcia, A. Effect of mycoparasitic fungi on the development of Sphaerotheca fusca in melon leaves. Mycological Research. 2003, 107, 64–71.
  4. Goettel, M.S.; Koike, M.; Kim, J.J.; Aiuchi, D.; Shinya, R.; Brodeur, J. Potential of Lecanicillium spp. for management of insects, nematodes and plant diseases. Journal of invertebrate Pathology. 2008, 98, 256–261.
  5. Dhingra, O.D.; Coelho-Netto, R.A.; Rodrigues, F.A., Silva, G.J.; Maia, C.B. Selection of endemic nonpathogenic endophytic Fusarium oxysporum from bean roots and rhizosphere competent fluorescent Pseudomonas species to suppress Fusarium-yellow of beans. Biological control. 2006, 39, 75–86.
  6. Singh, A.; Shukla, N.; Kabadwal, B.C.; Tewari, A.K.; Kumar, J. Review on plant Trichoderma pathogen interaction. International journal of current microbiology and applied sciences. 2018, 7, 2382–2397.
  7. Egel, D.S.; Hoagland, L.; Davis, J.; Marchino, C.; Bloomquistc, M. Efficacy of organic disease control products on common foliar diseases of tomato in field and greenhouse trials. Crop protection. 2019, 122, 90–97.
  8. Skinner, M.; Parker, B.L.; Kim, J.S. Role of entomopathogenic fungi in integrated pest management. Integrated Pest Management. 2014, 272–266.
  9. Rouphael, Y.; Franken, P.; Schneider, C.; Schwarzd, D.; Giovannetti, M.; Agnolucci, M.; De Pascale, S.; Bonini, P., Colla, G. Arbuscolar mychorrhizal fungi act as biostimulants in horticultural crops. Scientia Horticulturae. 2015, 196, 91–108.
  10. Mo, Y.; Wang, Y.; Yang, R.; Zheng, J.; Liu, C.; Li, H.; Ma, J.; Zhang, Y.; Wei, C.; Zhang, X. Regulation of plant growth, photosynthesis, antioxidation and osmosis by an arbuscolar mychorrhizal fungus in watermelon seedlings under well-watered and drought conditions. Frontiers in plant Science. 2016, 7, 644.
  11. Colla, G.; Rouphael, Y.; Di Mattia, E.; El-Nakhel, C.; Cardarelli, M. Co-inoculation of Glomus intraradices and Trichoderma atroviride acts as a biostimulant to promote growth, yield and nutrient uptake of vegetable crops. Journal of the Science of Food and Agriculture. 2015, 95, 1706–1715.
  12. Zhang, S.; Gan, Y.; Xu, B. Application of plant growth promoting fungi Trichoderma longibrachiatum T6 enhances tolerance of wheat to salt stress through improvement of antioxidative defense system and gene expression. Frontiers in Plant Science. 2016, 7, 1405.

Paragraphs 5 to 8 have been modified and expanded according to suggestions

  1. Preparation and application of commercial biofertilizers

Most commercial products contain Trichoderma as an active ingredient and some formulations contain several species belonging to this genus: T. asperellum, T. gamsii, T. viride, T. harzianum. A multitude of commercial proposals, with a predominantly bi-ostimulant function, have a mixed microbiological composition; rather widespread is the association of mycorrhizae of the genus Glomus with rhizosphere bacteria (Bacillus spp., Pseudomonas spp, Azotobacter spp., Azospirillum spp., Rhyzobium spp.) and Trichoderma spp. Other useful fungi sold in mixtures of mycorrhizal inocula belong to the genus Rhi-zophagus, Clonostachys, Arthrobotrys, Pochonia, Dactylella and yeasts of the genus Pichia. The combined use of Trichoderma harzianum with different strains of Bacillus subtilis in repeated pre- and post-transplant treatments can control tomato tracheofusariosis and stimulate both growth and biosynthesis of vitamin C and lycopene in the berries [78]. These two microorganisms were also combined with a strain of Pseudomonas fluorescens and vermicompost, producing the dual effect of reducing tomato tracheofusariasis and increasing antioxidant compounds in the berries [79]. Also in tomato, the synergistic effect of Trichoderma spp. and Pseudomonas fluorescens was observed in the biocontrol of bacterial wilt caused by Ralstonia solanacearum [80]. The joint use of Trichoderma, Bacillus and Pseudomonas, supported by compost, reduced the incidence of tracheofusariasis in lettuce grown in open fields by up to 69% [81]. The use of composted oak bark both reduced the ability of Trichoderma to contain Phytophthora infestans on tomato and enhanced the biocontrol efficacy of Bacillus subtilis [82]. On potato, the combined treatment of tubers with Bacillus subtilis and soil with a mixture of Trichoderma koningii and T. harzianum controlled R. solani and stimulated vegetative plant growth [83].

References

  1. Alekseeva, K.L.; Smetanina, L.G.; Kornev, A.V. Biological protection of tomato from Fusarium wilt. AIP Conference Proceedings. 2019, 2063, 30001
  2. Basco, M.J.; Bisen, K.; Keswani, C.; Singh, H.B. Biological management of Fusarium wilt of tomato using biofortified vermicompost. Mycosphere. 2017, 8, 467–483.
  3. Yendyo, S.; Ramesh, G.C.; Pandey, B.R. Evaluation of Trichoderma spp., Pseudomonas fluorescens and Bacillus subtilis for biological control of Ralstonia wilt of tomato. F1000Research. 2018, 6, 2028.
  4. Cucu, M.A.; Gilardi, G.; Pugliese, M.; Matic, S.; Gisi, U.; Gullino, M.L., Garibaldi, A. Influence of different biological control agents and compost on total and nitrification driven microbial communities at rhizosphere and soil level in a lettuce – Fusarium oxysporum f. sp. Lactucae pathosystem. Journal of Applied Microbiology. 2019, 126, 905–918
  5. Bahramisharif, A.; Rose, L.E. Efficacy of biological agents and compost on growth and resistance of tomatoes to late blight. Planta. 2019, 249, 799–813.
  6. Abeer, A.A.; Abd El-Kader, A.E.S.; Ghoneem, K.H.M. Two Trichoderma species and Bacillus subtilis as biological control agents Rhizoctonia disease and their influence in potato producivity. Egyptian Journal of Agricultural Research. 2017, 95, 527–541.

  1. Formulated biofertilizers: application methods

The inoculation of fungi can take place by direct administration of spores or mycelium fragments. Numerous formulations are marketed as wettable powders, pastes, creams, water-dispersible microgranules, pellets or liquid preparations. It is essential to comply with the recommended dose and mode of administration stated on the label, and to take into account the expiry date and storage conditions of the product. The presence of chemical residues in the soil and on the crop and the subsequent application of other sterilising treatments may limit the viability and development of beneficial fungi, com-promising the effectiveness of the micro-organism treatment. In order to ensure the sur-vival of fungal inocula, enhance saprophytic capacities and encourage colonisation of the rhizosphere, it is advisable to maintain a temperature and pH range suitable for vegetative development, a good supply of organic matter in pre-biotic soils and to exclude de-structive chemical treatments. Treatment is more effective if an initial application is made at the highest dose and repeated applications are made even at lower concentrations; the possibility of increasing the frequency of treatments improves efficacy. Beneficial fungi are used for preventive purposes except in cases where the presence of the pathogen is necessary to allow it to take root and guarantee efficacy. The functionality of the consor-tium is not always guaranteed by the number of microorganisms: it is essential to seek compatibility and synergies between individuals. For example, in the biocontrol of Rhi-zoctonia solani in bean by evaluating different combinations and inoculation times of Trichoderma harzianum, Rhizophagus intraradices and Bacillus pumilus, it emerged that in simultaneous treatment with substrate infection, the best combination in terms of disease reduction was shown by the Bacillus - Trichoderma combination. In prevention, on the other hand, good control was achieved with Trichoderma alone, while the combination T. harzianum - R. intraradices had no significant effect [95]. On soybean a consortium con-sisting of Trichoderma citrinoviride, Pseudomonas aeruginosa, Bacillus cereus and Bacillus am-yloliquefaciens was tested against Macrophomina phaseolina and Sclerotinia sclerotiorum [96]. The combination of microorganisms was most active in the production of ammonium, siderophores and lytic enzymes. The consortium consisting of Trichoderma harzianum, Epicoccum spp., Bacillus megatherium and B. amyloliquefaciens was successfully employed for the control of black spot in the caryopsis of wheat, caused by the Cochliobolus sativus complex, Alternaria alternata and Fusarium graminearum. In the field, the microbial con-sortium increased germination and tillering, reduced the incidence of leaf spot and in-creased seed weight [88].

The combination of Trichoderma harzianum- Pseudomonas fluorescens had a synergistic effect in the biocontrol of rice bruson, caused by Magnaporthe oryzae, and leaf blight due to the bacterium Xanthomonas orza pv. Oryzae [97]. On tree species, the combination of avirulent strains of Fusarium oxysporum, Phoma sp. and Pseudomonas fluorescens has the ability to reduce the aggressiveness of Verticillium dahlie attacks [98].

References

  1. Hussein, A.N.; Abbasi, S.; Sharifi, R.; Jamali, S. The effect of biocontrol agents consortia against Rhizoctonia root rot of common bean Phaseolus vulgaris. Journal of crop Protection. 2018, 7, 73–-85.
  2. Thakkar, A.; Saraf, M. Development of microbial consortia as a biocontrol agent for effective management of fungal diseases in Glycine max L. Archive of Phytopathology and Plant Protection. 2015, 48, 459–-474.
  3. Jambhulkar, P.P.; Sharma, P.; Manokaran, R.; Lakshman, D.K.; Rokadia, P.; Jambhulkar, N. Assessing synergism of combined applications of Trichoderma harzianum and Pseudomonas fluorescens to control blast and bacterial leaf blight of rice. European Journal of plant Pathology. 2018, 152, 747–-757.
  4. Varo, A.; Raya-Ortega, M.C.; trapero, A. Selection and evaluation of micro-organisms for biocontrol of Verticillum dahlie in olive. Journal of Applied Microbiology. 2016, 121, 766–-767.

Reviewer 3 Report

This review is suggested to cover quite a broad topic of the role of microbial fertilizers in crop production. In general, many aspects of this problem are being discussed, for example mode of action of rhizobacteria, plant defense mechanisms, induced stress tolerance in plants and commercial applications of biofertilizers. However, the suggested article topic is too broad, and from my point of view many of the questions were not covered in this review. Also, the review structure should be improved, the information could be structured better. I suggest changing the article name to something more specific like rhizobacteria-based fertilizers.

Line 13: “improve soil nutrients” should it be “improve plant nutrients' status” or “increase soil nutrient content”?

Line 17-18 The microbial inoculation couldn’t be a substitute for the mineral fertilizers. The use of organic fertilizers is also eco-friendly.

Lines 18-19 “plant growth and soil fertility” is repeated twice in one sentence.

Line 23 Also repetitive

Line 41 “Antibiotics, parasitism, and suppression of rhizobacterial growth” it should be clarified how these aspects could improve plant growth.

Line 79 phytopathogenic microorganisms are not capable for all that, that’s exaggeration 

Line 100 “Rhizobacteria produce these phytohormones.” It would be great to add information on how much (or percentage) of phytohormones are produced by a plant itself and by rhizobacteria.

Line 105 phytopathogens are mentioned twice, please check.

Line 121 IAA – the abbreviation is mentioned for the first time, it’s necessary to explain it.

Line 133 I would recommend splitting the paragraph here, and to add more information regarding P solubilization. 

Line 139 Table 2 what was the nature of long-term stress? Please explain.

Line 152 – please check again the types of plant resistances to pathogens and write this part more clearly

Line 157 – again, the information regarding iron should be placed separately. Maybe it would be nice to make a separate chapter for nutrients that could be provided by PGPR.

Line 175 misspelling in the word “hydrogen”

Line 183, Figure 2. There is a misspelling, and the figure does not clearly inform the potential reader what is affecting on what.

Line 208 from my opinion, liquid formulations are less stable than inoculants on solid carriers. Could you please provide some proof of your statement?

Line 222 “It sells” please clarify

Line 215, the part 6 – this part is not well structured. I would suggest rewriting it to make it clearer 

Line 285 regarding Zn – maybe biofertilizers could transform unavailable forms of Zn to more available ones? Otherwise, there is little help of them to plant growth, and they cannot substitute chemical fertilizers in the providing plants by Zn

Lines 296 -300 this is too general

Part 8 – this information is well known and too general, it would be great to see any examples of real applications.

Conclusion covers some moments that were not discussed in the article (the myths for example)

The language of the article should be improved. I recommend spell-check

Author Response

Responses to the third reviewer

Good evening I am sending you the changes made according to your suggestions. The article was also checked by an English-speaking scientist.

This review is suggested to cover quite a broad topic of the role of microbial fertilizers in crop production. In general, many aspects of this problem are being discussed, for example mode of action of rhizobacteria, plant defense mechanisms, induced stress tolerance in plants and commercial applications of biofertilizers. However, the suggested article topic is too broad, and from my point of view many of the questions were not covered in this review. Also, the review structure should be improved, the information could be structured better. I suggest changing the article name to something more specific like rhizobacteria-based fertilizers.

Line 13: “improve soil nutrients” should it be “improve plant nutrients' status” or “increase soil nutrient content”?

I have replaced the sentence as suggested

Line 17-18 The microbial inoculation couldn’t be a substitute for the mineral fertilizers. The use of organic fertilizers is also eco-friendly.

I have replaced the sentence as suggested

Therefore, it is believed that the use of microbes as bioinoculants, used together with chemical fertilisers, is the best strategy to increase plant growth and soil fertility.

Lines 18-19 “plant growth and soil fertility” is repeated twice in one sentence.

I have replaced the sentence as suggested

Line 23 Also repetitive

I have replaced the sentence as suggested

Line 41 “Antibiotics, parasitism, and suppression of rhizobacterial growth” it should be clarified how these aspects could improve plant growth.

I have replaced the sentence as suggested

Furthermore, through induced systemic resistance (SRI), competition with nutrients, antibiotics, parasitism and growth suppression of rhizobacteria are mechanisms that lead to increased plant resistance [6].

Line 79 phytopathogenic microorganisms are not capable for all that, that’s exaggeration

I have replaced the sentence as suggested

Sustainable agriculture and plant cultivation can be threatened by the presence of microorganisms, with a deterioration in plant quality and production yields [24].

Line 100 “Rhizobacteria produce these phytohormones.” It would be great to add information on how much (or percentage) of phytohormones are produced by a plant itself and by rhizobacteria.

I have replaced the sentence as suggested

Several naturally occurring auxin-like molecules have been described as products of bacterial metabolism in Azospirillum sp. cultures. In addition to Indole-3-acetic acid (IAA) (between 5 and 50 lg ml-1 typically produced according to culture conditions and strain), Indol-3-butirric acid (IBA) [30], and Phenylacetic acid (PAA) [31], considered in sensu stricto as real auxins, many other indolic compounds (precursors and/or catabolites) have been identified in Azospirillum sp. supernatants, including indole-3-lactic acid (ILA), indole-3-ethanol and indole-3-methanol, indole-3-acetamide (IAM) [32], indole-3-acetaldehyde [33], tryptamine (TAM), anthranilate, and other uncharacterized indolic compounds [34].

Line 105 phytopathogens are mentioned twice, please check.

I have replaced the sentence as suggested

Line 121 IAA – the abbreviation is mentioned for the first time, it’s necessary to explain it.

I have replaced the sentence as suggested

Line 133 I would recommend splitting the paragraph here, and to add more information regarding P solubilization.

A new paragraph has been inserted

  1. Microorganisms that solubilize phosphate

There are large quantities of phosphate in soil, but they are in an insoluble form that plants cannot utilise for growth since they are insoluble [45]. A group of organisms known as phosphate-solubilizing microorganisms (PSMs) consists of actinobacteria, bacteria, fungi, arbuscular mycorrhizae, and cyanobacteria that are capable of hydrolyzing organic and inorganic phosphorus into soluble forms, making it available to plants. In Indonesia, Djuuna et al. [46] sampled soil microorganisms, which are commonly associated with the rhizosphere [47]. Agricultural soils with a relevant history of growing vegetables, cereals, and legumes from different regions were collected. The results showed a population of solubilizing bacteria ranging between 25 × 103 and 550 × 103 CFU g–1 of soil and solubilizing fungi between 2.0 × 103 and 5.0 × 103 CFU g–1 of soil in all areas examined. There is also great diversity in PSM. It is known that bacteria belong to the genera Azospirillum, Bacillus, Pseudomonas, Nitrosomonas, Erwinia, Serratia, Rhizobium, Xanthomonas, Enterobacter, and Pantoea [47,48]. Non-mycorrhizal fungi include Penicillium, Fusarium, Aspergillus, Alternaria, Helminthosporium, Arthrobotrys, and Trichoderma, [62,65]. Rhizophagus irregularis [49,50], Glomus mossea, G. fasciculatum, and Entrophospora colombiana are examples of mycorrhizal fungi. PSM occurs in actinobacteria such as Streptomyces, Thermobifida, and Micrococcus [51–54], as well as cyanobacteria including Calothrix braunii, Westiellopsis prolifica, Anabaena variabilis, and Scytonema sp.

Line 139 Table 2 what was the nature of long-term stress? Please explain.

High temperature has been entered

Line 152 – please check again the types of plant resistances to pathogens and write this part more clearly

I have replaced the sentence as suggested

These include cellulases, chitinases, lipases and proteases secreted by the plant. Plants respond to pathogens in two ways: acquired systemic resistance (SAR) and induced systemic resistance (ISR). SAR is implemented in response to a pathogen pre-infection, inducing a hypersensitive reaction, recognisable by a local necrotic lesion of the tissue and an accumulation in the cells of salicylic acid (SA). ISR, on the other hand, induces no visible symptoms and the cells rarely contain (SA) [71-74].

Line 157 – again, the information regarding iron should be placed separately. Maybe it would be nice to make a separate chapter for nutrients that could be provided by PGPR.

A new paragraph has been inserted

Plants require a small amount of iron from the earth's crust, but Fe deficiency is a nutritional disorder caused by a lack of iron. Plants and microorganisms cannot easily utilize this nutrient in soil because the forms it finds are usually Fe3+ oxy-hydroxides. For Fe3+ to be readily consumed by plants and microorganisms, it must be reduced to Fe2+ [55–57]. Several soil microorganisms have been shown to play a critical role in diminishing Fe deficiency as an environmentally-friendly alternative agricultural practice. As well as al-leviating biotic and abiotic stresses, these microorganisms have been shown to be benefi-cial [58,59]. There are rhizobacteria that can colonize the rhizosphere environment, some of which promote nutrient uptake and plant growth; hence, they are referred to as plant growth-promoting rhizobacteria (PGPR) [60,61]. According to their relationship with plant roots, PGPRs fall into two groups: i) extracellular PGPRs inhabit the rhizosphere, or spaces between root cortex cells, and ii) intracellular PGPRs inhabit root cells specialized in leguminous nodules [62]. Micrococcus, Pseudomonas, Agrobacterium, and Bacillus are some of the extracellular PGPR genera. Several studies have shown that PGPR can en-hance Fe uptake under limited Fe availability conditions by accumulating and exuding organic acids, phenolic compounds, siderophores and enhancing FCR enzyme activities, in cucumber [63], Arabidopsis [64], pear [65], peach [66], and apple rootstocks [67]. The beneficial effects of PGPR on Fe deficiency have been demonstrated in several studies, but few studies have explored the molecular mechanisms by which PGPR enhances plant Fe uptake. As a result, Zhou et al. [64] and Aras et al. [67] have reported that PGPR activates iron deficiency-related genes like ferric chelate reductase (FRO2) and Fe2+ transporter (IRT1).

Line 175 misspelling in the word “hydrogen”

Correction has been made

Line 183, Figure 2. There is a misspelling, and the figure does not clearly inform the potential reader what is affecting on what.

Figure 2 has been modified as also suggested by other reviewers

Line 208 from my opinion, liquid formulations are less stable than inoculants on solid carriers. Could you please provide some proof of your statement?

I deleted the part about stability in the text because liquid solutions are actually less stable than solid ones.

Line 222 “It sells” please clarify

I have replaced the sentence as suggested

Azospirillum inoculants can be found in Europe and South Africa, where a number of products, including barley, maize, sorghum and wheat, pre-inoculated with Azospirillum brasilense are already marketed.

Line 215, the part 6 – this part is not well structured. I would suggest rewriting it to make it clearer

Paragraph 6 has been modified as also suggested by other reviewers

Line 285 regarding Zn – maybe biofertilizers could transform unavailable forms of Zn to more available ones? Otherwise, there is little help of them to plant growth, and they cannot substitute chemical fertilizers in the providing plants by Zn

I have replaced the sentence as suggested

In developing countries, cereals are a major source of calories, but they are also low in zinc because they are mostly grown in soils lacking it. Health problems related to zinc deficiency can result from cereal-based diets. The application of microbial biofertilisers can transform poorly available forms of zinc into more available and absorbable forms for plants.

Lines 296 -300 this is too general

I have replaced the sentence as suggested

An essential and safe method for increasing plant growth, resistance to biotic and abiotic stresses and increasing product quality is the use of microbial biofertilisers. In terms of increasing productivity, it is a promising solution [142,143].

Part 8 – this information is well known and too general, it would be great to see any examples of real applications.

I have replaced the sentence as suggested

Microorganisms can significantly improve Fe accumulation in wheat in an efficient and eco-friendly way. Strains of Bacillus spp. form spores and are widely explored as plant growth promoting bacteria (PGPB) in contemporary agriculture for different purposes [137,138]. They secrete siderophores, organic acids, and other compounds to promote the uptake of Fe in the rhizosphere of wheat [139,140]. Several Bacillus and Paenibacillus species increase P, N, K, Fe and Zn contents in maize [141].

Conclusion covers some moments that were not discussed in the article (the myths for example)

I edited the part on myths as suggested

For the widespread use of microbial bio-fertilisers, it would be useful to inform farmers on how best to use them, how to store them, the benefits they can bring to plant cultivation, the possibility of being able to reduce fertilisers and pesticides, and the safety for the operator in application

Round 2

Reviewer 2 Report

The manuscript has been improved.The authors have contributed new sections to the manuscript. However, the abstract and introduction do not take these new contributions into account.

The word BIOFERTILISERS must be used throughout the manuscript.

Author Response

Responses to the second reviewer

Good evening I am sending you the changes made according to your suggestions. The article was also checked by an English-speaking scientist.

The manuscript has been improved.The authors have contributed new sections to the manuscript. However, the abstract and introduction do not take these new contributions into account.

The word BIOFERTILISERS must be used throughout the manuscript.

 The missing part has been added to both the abstract and the introduction. In addition, all the terms Biofertilizers have been corrected in the text

Reviewer 3 Report

Dear authors, thank you for the response to the previous comments. Here are new comments that appeared while reading the revised version:

Line 212 – 218 – please check the structure of these sentences, some parts are lacking.

Line 239 – genera names should be in italic.

Line 242 – what is FCR? This abbreviation should be deciphered.

Line 367 – what is zero-residue? If it is zero-waste, is it applicable to agricultural production?

Lines 375, 389 - telluric – shall it be terrestrial?

Line 413 – the things are a bit mixed here: “to promote the structure of soil aggregates, improving their physical fertility”. Yes, AM fungi contribute to soil physical stability, however, there is no direct link with the soil fertility. Please check

Lines 550, 553 – “by direct administration” – maybe direct application?

Line 620, 621 – the correction has been made, but it’s necessary to provide a link to prove this statement. Otherwise it would be better to rewrite the part about Zn

it is fine

Author Response

Responses to the third reviewer

Good evening I am sending you the changes made according to your suggestions. The article was also checked by an English-speaking scientist.

Dear authors, thank you for the response to the previous comments. Here are new comments that appeared while reading the revised version:

Line 212 – 218 – please check the structure of these sentences, some parts are lacking.

The entire deficient part has been corrected and expanded

Line 239 – genera names should be in italic.

All genre names have been put in italics

Line 242 – what is FCR? This abbreviation should be deciphered.

The term has been described and corrected

Line 367 – what is zero-residue? If it is zero-waste, is it applicable to agricultural production?

The whole sentence has been corrected and explained

Lines 375, 389 - telluric – shall it be terrestrial?

The term has been replaced with the correct one

Line 413 – the things are a bit mixed here: “to promote the structure of soil aggregates, improving their physical fertility”. Yes, AM fungi contribute to soil physical stability, however, there is no direct link with the soil fertility. Please check

The sentence has been corrected and described more appropriately

Lines 550, 553 – “by direct administration” – maybe direct application?

The term has been replaced with the correct one

Line 620, 621 – the correction has been made, but it’s necessary to provide a link to prove this statement. Otherwise it would be better to rewrite the part about Zn

The correct bibliography has been inserted next to Zn
